# OpenReview forum: "RAG "Hype" vs. Reality"
_ICLR.cc/2026/Conference — Submitted to ICLR 2026_

### Official Review · Reviewer_oPKX · 2025-10-25

**Soundness:** 2
**Presentation:** 3
**Contribution:** 1
**Rating:** 2
**Confidence:** 4

**Summary:**

This paper argues that while Retrieval-Augmented Generation (RAG) offers practical benefits for accessing up-to-date information, it is limited by retrieval quality, superficial knowledge integration, and poor support for complex reasoning. In contrast, finetuning enables deeper, more robust knowledge assimilation and superior long-context understanding, making it a more powerful and sustainable path toward advanced, specialized LLMs.

**Strengths:**

* The paper is well written
* Discussing the necessity of RAG is an interesting topic

**Weaknesses:**

* The paper tells several weakness of RAG, but most are well known in the community
* The paper claims that RAG fails to help reasoning among contexts while finetuning could, but experimental validation is needed
* The authors claim that the efficiency problem of finetuning can be solved by PEFT, but simply conducting SFT will lead to catastrophic forgetting, which greatly influence the performance on out domain datas.

**Questions:**

* The authors claim that LLMs exhibit difficulty in effectively utilizing information spread across very long contexts, However, current LLMs like Qwen Llama have already support context of 256k, why you still thinks this is still a problem, this should have been solved during the training of Base/Instruct Model
* SFT can indeed help the model better understand the context. But it is at cost of catastrophic forgetting, but the paper does not discuss the problem
* In my opinion, SFT and RAG are quite different, SFT can help the model to conduct better reasoning, but it does not conflict with retrieval relevant knowledge in the context, SFT can help the model for better RAG instead of directly replacing it.

---

### Official Review · Reviewer_XML7 · 2025-10-28

**Soundness:** 1
**Presentation:** 2
**Contribution:** 1
**Rating:** 2
**Confidence:** 4

**Summary:**

The paper examines the gap between the hype and the practical reality of RAG, evaluating when and why RAG delivers measurable gains. The authors argue that RAG, despite its popularity and practical utility, is fundamentally limited by its reliance on discrete retrieval  steps, leading to superficial knowledge integration, retrieval bottlenecks, and challenges in long-context reasoning. In contrast, they position finetuning as the superior approach for achieving deep knowledge assimilation and true long-context understanding through parametric adaptation.

However, this paper suffers from severe methodological and scholarly deficiencies that undermine its credibility. Most critically, the authors present strong comparative claims, asserting that one approach is fundamentally superior to another without conducting any original experiments or systematic empirical validation. This is not a position paper grounded in evidence; it is an opinion piece dressed in academic language. The paper cherry-picks citations from blog posts, medium articles, and industry commentary to support a predetermined conclusion, while ignoring substantial peer-reviewed research that contradicts its thesis.

In summary, this submission fails to meet the standards expected of a premier machine learning conference. It reads more as advocacy or marketing material than as rigorous academic research, as it lacks scientific rigor, empirical validation, balanced analysis, and novel insights. The paper would be more appropriately published as a blog post or opinion piece rather than subjected to peer review at ICLR.

**Strengths:**

. The paper is well-organized with a clear argumentative structure. The progression from RAG critique to finetuning advocacy to comparative analysis is logical and easy to follow.
2. The paper provides a thorough enumeration of RAG's failure modes. It provides multi-dimensional perspective spanning retrieval quality, coverage, prompting, and cost.

**Weaknesses:**

1. The manuscript advances strong comparative claims without presenting original experiments, benchmarks, or systematic evaluations. The manuscript provides no controlled comparison demonstrating that finetuning outperforms RAG on identical tasks with identical data. Citations to existing studies are selective and often anecdotal, as illustrated by the reported 49.85% improvement for medical Q&A in Section 4.5.
2. The critique addresses only a basic view of RAG while overlooking advanced techniques. Iterative retrieval approaches, including Self-RAG, Corrective RAG, and Adaptive RAG, refine retrieval through repeated feedback. Hybrid retrieval methods combine dense and sparse retrieval, for example pairing BM25 with vector search. Query reformulation and expansion directly target retrieval bottlenecks. Learned retrievers are trained for specific downstream tasks. These approaches address many of the limitations listed in Sections 2.2 to 2.4, yet they are not considered in the manuscript.
3. The treatment of finetuning’s limitations is brief and insufficient. Although Section 3.4 notes some challenges, these issues are quickly minimized as mere engineering concerns without supporting analysis.
4. The manuscript frames RAG and finetuning as competing approaches, although in practice they are complementary. Many successful systems combine both methods, a point briefly acknowledged in Section 4.3 but not examined in depth. Different use cases legitimately favor different approaches. The “cornerstone” framing in the title and conclusion is overly strong and insufficiently supported.
5. The manuscript relies heavily on non-peer-reviewed sources, including blogs and industry commentary. While contemporary perspectives can be informative, the limited use of rigorous peer-reviewed evidence weakens the scientific credibility of the work.
6. The presentation lacks visual and quantitative support. The manuscript does not include figures contrasting RAG and finetuning architectures, tables summarizing performance, cost, and latency, empirical plots derived from the literature, or decision frameworks to guide method selection.
7. The claim about long-context understanding is insufficiently justified. Section 3.3 asserts that finetuning is crucial for unlocking true long-context capability, yet it does not explain a mechanism by which finetuning achieves this outcome. The alleged weakness of RAG and superiority of finetuning in long-context settings is not empirically validated. Recent work on native long-context models with million-token windows receives limited attention.
8. The discussion of hybrid methods is limited and prematurely dismissive. Section 4.3 mentions RAFT and related hybrids yet suggests that their mere existence supports the thesis, which oversimplifies the issue. Hybrid approaches may in fact be the optimal solution in many settings, and their architectural and operational complexity deserves careful analysis.

**Questions:**

1. Can you provide controlled experimental comparisons between RAG and finetuning on standardized benchmarks such as question answering, summarization, and domain-specific tasks? In the absence of such evidence, on what basis do you justify claims of “deeper knowledge integration” and “superior performance,” beyond assertion?
2. How do your critiques apply to state-of-the-art RAG systems that use iterative retrieval, query reformulation, hybrid dense-sparse retrieval, or learned retrievers? Do your arguments actually engage these methods, or do those techniques eliminate the very limitations you claim exist, rendering the critique outdated?
3. You acknowledge that finetuning requires high-quality training data yet dismiss this as an “engineering and resource problem.” In domains where such data do not exist or are prohibitively expensive to create, on what grounds is finetuning the “cornerstone” solution? Provide concrete guidance on data scale, quality thresholds, collection/labeling protocols, and feasibility.
4. Section 3.4 mentions catastrophic forgetting but does not quantify its severity. In continual-learning settings where models must incorporate new information while retaining general capabilities, how does finetuning compare to RAG’s capacity for simple knowledge-base updates? Where is the evidence that finetuning preserves prior competence while integrating new knowledge?
5. Can you provide a quantitative cost-benefit analysis comparing RAG and finetuning  across different scales (small startups vs. large enterprises) and use cases (dynamic information retrieval vs. static domain expertise)？
6. What is the concrete mechanism by which finetuning purportedly enables better long-context understanding than supplying the same context via RAG? Cite specific studies and benchmarks demonstrating finetuned models outperforming RAG in long-context regimes; otherwise the claim reads as marketing rather than science.
7. If hybrid RAG-plus-finetuning approaches (e.g., RAFT) achieve superior results relative to either method alone, does this not undercut the thesis that finetuning is the “cornerstone”? Should the conclusion instead be that hybrid methods constitute the optimal path in practice?
8. Provide real-world production case studies where pure finetuning has successfully replaced RAG for complex, rapidly changing knowledge tasks, including the application domain, workload characteristics, evaluation metrics, update cadence, and operational timelines.
9. Why does the manuscript rely heavily on blogs and industry commentary rather than peer-reviewed research? Can each central claim be substantiated with rigorous academic sources, and where your position deviates from the literature, what evidence justifies that departure?

---

### Official Review · Reviewer_68BP · 2025-11-01

**Soundness:** 2
**Presentation:** 2
**Contribution:** 2
**Rating:** 2
**Confidence:** 2

**Summary:**

This position paper challenges the dominant narrative around Retrieval-Augmented Generation (RAG), arguing that fine-tuning is superior for building specialized, knowledgeable LLMs. The authors contend that RAG is overhyped with fundamental architectural limitations, while fine-tuning offers deeper knowledge integration and better long-context understanding.

**Strengths:**

1. Timely and Provocative Thesis
The paper addresses a highly relevant debate in the LLM community, challenging the widespread adoption of RAG systems. The contrarian stance generates valuable discussion about technology choices beyond marketing hype.
2. Comprehensive Critique Framework
The authors systematically identify RAG's limitations across multiple dimensions: retrieval bottleneck, superficial knowledge integration, long-context illusions, latency overhead, and vector database hype. This multi-faceted analysis provides a structured way to think about RAG's weaknesses.
3. Practical Considerations
The paper highlights often-overlooked operational complexities of RAG systems (data curation, indexing maintenance, retrieval service management) and acknowledges PEFT techniques that make fine-tuning more accessible.

**Weaknesses:**

1. Lack of Empirical Evidence
As a position paper, it relies heavily on citing others' work without original experiments. Critical claims like "fine-tuning is essential for long-context mastery" lack systematic empirical validation. No controlled comparisons across diverse benchmarks are provided.
2. Overly Binary Framing
The paper oversimplifies a nuanced technical decision into "RAG bad, fine-tuning good." It dismisses RAG's core strengths (dynamic updates, verifiable sourcing, lower barrier to entry) as mere "tactical" advantages while framing fine-tuning's significant challenges (catastrophic forgetting, data requirements, computational costs) as "addressable engineering problems."
3. Strawman Argumentation
The paper attacks an exaggerated version of RAG advocacy that few serious researchers actually hold. Most practitioners already recognize RAG limitations and use hybrid approaches. The "hype" being criticized stems more from commercial marketing than academic consensus.
4. Questionable Claims About Long Context
The assertion that fine-tuning "unlocks" long-context capabilities is speculative. The "lost in the middle" problem is an architectural limitation of transformer attention, which fine-tuning alone may not fundamentally solve.
5. Insufficient Treatment of Use Cases
The paper inadequately addresses scenarios where RAG is objectively superior: rapidly changing knowledge bases (news, regulations), strict source attribution requirements (legal, medical), and resource-constrained environments. The dismissal of these as "niches" underestimates their real-world prevalence.
6. Misinterpretation of Hybrid Methods
The emergence of RAFT and similar techniques is presented as evidence of RAG's inadequacy, when it more accurately demonstrates the complementary nature of both approaches.

**Questions:**

Can you elaborate more on the innovation point of this paper?

---

### Official Review · Reviewer_gnkf · 2025-11-02

**Soundness:** 1
**Presentation:** 2
**Contribution:** 1
**Rating:** 2
**Confidence:** 4

**Summary:**

This position paper argues that finetuning—rather than overhyped Retrieval-Augmented Generation (RAG)—is the cornerstone for enhancing LLMs’ deep knowledge integration and long-context capabilities: it critiques RAG for inherent limitations like retrieval quality dependence, superficial knowledge integration, long-context "lost in the middle" issues, and overstated vector database importance, while highlighting finetuning’s strengths in parametric adaptation; however, the paper has some flaws, including over-reliance on existing literature without original empirical data, one-sided dismissal of recent RAG advancements, insufficient exploration of hybrid approaches, and downplaying finetuning’s practical barriers (e.g., high costs, data scarcity) as trivial.

**Strengths:**

1. The paper makes a distinct position on the comparative value of RAG and finetuning, providing a focused critique of RAG’s limitations and advocating for finetuning as a more foundational approach.
2. The paper covers multiple aspects, including knowledge integration depth, long context handling, robustness, and operational trade-offs, offering a holistic comparison of the two paradigms.
3. The paper incorporates recent research papers and industry discussions to support the claims.

**Weaknesses:**

1. Although this is a position paper, the argument is only based on qualitative analysis and existing papers, lacking original empirical data to validate claims about finetuning’s superiority.
2. Some criticisms of RAG (e.g., retrieval bottlenecks) overlook recent advancements in RAG systems or improved retrieval algorithms, presenting a somewhat one-sided view.
3. While hybrid approaches are mentioned in the paper, authors do not explore the potential in detail, missing an opportunity to address how RAG and finetuning can complement each other effectively.
4. The paper downplays finetuning’s practical barriers (e.g., data scarcity, catastrophic forgetting) as "addressable engineering issues" without sufficient discussion of their real-world impact.

**Questions:**

1. Some original empirical results in the position paper is needed, to compare the performance of finetuned models, RAG systems, and hybrid approaches on complex reasoning tasks requiring deep domain knowledge.
2. Please review the recent advancements in RAG that aim to mitigate the retrieval bottleneck and superficial knowledge integration issues the authors highlight in this paper.
3. The authors acknowledges finetuning’s high computational costs and data curation demands but treats them as trivial, it would be more practical to offer targeted recommendations for small and medium-sized organizations (which lack large computing resources and high-quality datasets) on how to implement finetuning for deep knowledge integration, rather than only emphasizing finetuning’s theoretical advantages.
4. The paper claims finetuning is superior to RAG in addressing the "lost in the middle" effect in long-context processing, but provides no specific mechanisms or empirical evidence to support this.

---

### Meta-Review · Area_Chair_W7cp · 2026-01-07

**Summary:**

This position paper contends that finetuning is the superior approach for knowledge integration, positioning RAG as an overhyped, tactical stopgap. While the reviewers found the premise timely and the critique of RAG's limitations structured, the submission was unanimously rejected. The consensus is that the paper fails as a scientific contribution due to a total lack of original empirical validation. It presents strong assertions as facts without experimental backing, relies heavily on non-peer-reviewed sources, and sets up a false dichotomy between RAG and finetuning while ignoring modern advancements in hybrid approaches.

**Reviewer Concerns:**

Lack of Empirical Evidence (All Reviewers): The most critical flaw. The paper makes comparative performance claims (e.g., finetuning is better for reasoning) without a single controlled experiment, benchmark, or data point to prove it.

One-Sided Argumentation (Reviewers XML7, 68BP): The paper critiques a "strawman" version of basic RAG, ignoring significant recent advancements (iterative RAG, GraphRAG, etc.) that solve many of the listed limitations.

False Dichotomy (Reviewers oPKX, XML7): The premise treats RAG and finetuning as mutually exclusive competitors, whereas the community largely views them as complementary. The dismissal of hybrid methods (like RAFT) is seen as premature.

Speculative Technical Claims (Reviewers 68BP, oPKX): Claims that finetuning inherently solves the "lost-in-the-middle" phenomenon or long-context reasoning issues are presented without mechanistic explanation or proof.

Practicality Issues (Reviewer gnkf): The paper waves away the massive costs and data curation challenges of finetuning as mere "engineering problems," which reviewers found unrealistic.

**Reviewer Scores:**

Reviewer gnkf: 2

Reviewer 68BP: 2

Reviewer XML7: 2

Reviewer oPKX: 2

---

### Decision · Program_Chairs · 2026-01-26

Reject